# Comparative Genomic Analysis of the Marine Cyanobacterium *Acaryochloris* *marina* MBIC10699 Reveals the Impact of Phycobiliprotein Reacquisition and the Diversity of *Acaryochloris* Plasmids

**DOI:** 10.3390/microorganisms10071374

**Published:** 2022-07-07

**Authors:** Haruki Yamamoto, Kazuma Uesaka, Yuki Tsuzuki, Hisanori Yamakawa, Shigeru Itoh, Yuichi Fujita

**Affiliations:** 1Graduate School of Bioagricultural Sciences, Nagoya University, Nagoya 464-8601, Japan; duffko0614@gmail.com (Y.T.); yamakawa008@gmail.com (H.Y.); fujita@agr.nagoya-u.ac.jp (Y.F.); 2Center for Gene Research, Nagoya University, Nagoya 464-8601, Japan; uesaka.kazuma.p1@f.mail.nagoya-u.ac.jp; 3Graduate School of Sciences, Nagoya University, Nagoya 464-8601, Japan; itoh@bio.phys.nagoya-u.ac.jp

**Keywords:** cyanobacteria, *Acaryochloris*, comparative genome analysis

## Abstract

*Acaryochloris* is a marine cyanobacterium that synthesizes chlorophyll *d*, a unique chlorophyll that absorbs far-red lights. *Acaryochloris* is also characterized by the loss of phycobiliprotein (PBP), a photosynthetic antenna specific to cyanobacteria; however, only the type-strain *A. marina* MBIC11017 retains PBP, suggesting that PBP-related genes were reacquired through horizontal gene transfer (HGT). *Acaryochloris* is thought to have adapted to various environments through its huge genome size and the genes acquired through HGT; however, genomic information on *Acaryochloris* is limited. In this study, we report the complete genome sequence of *A.* *marina* MBIC10699, which was isolated from the same area of ocean as *A. marina* MBIC11017 as a PBP-less strain. The genome of *A.*
*marina* MBIC10699 consists of a 6.4 Mb chromosome and four large plasmids totaling about 7.6 Mb, and the phylogenic analysis shows that *A.*
*marina* MBIC10699 is the most closely related to *A. marina* MBIC11017 among the *Acaryochloris* species reported so far. Compared with *A. marina* MBIC11017, the chromosomal genes are highly conserved between them, while the genes encoded in the plasmids are significantly diverse. Comparing these genomes provides clues as to how the genes for PBPs were reacquired and what changes occurred in the genes for photosystems during evolution.

## 1. Introduction

Photosynthetic organisms on earth convert solar energy into chemical energy that can be used by living organisms. Many photosynthetic organisms, including land plants, use chlorophyll *a* (Chl *a*) as the major photosynthetic pigment. Additionally, there are several derivatives with different substituents, such as Chl *b*, *c*, *d*, and *f*, and each has an absorption spectrum different from that of Chl *a* [1]. Those Chl derivatives are auxiliary pigments to Chl *a*, and in higher plants with Chl *a* and Chl *b*, only Chl *a* functions as the reaction center Chl to ignite the photosynthetic electron transfer, while Chl *b* does as antenna Chl in the LHC and other complexes [2]. The marine cyanobacterium *Acaryochloris marina* is a unique cyanobacterium that synthesizes Chl *d* as its primary photosynthetic pigment. *Acaryochloris* retains both Chl *d* and Chl *a*, but more than 90% of its total Chl is composed of Chl *d*, and the Chl special pair in the photosynthetic reaction centers is also occupied by Chl *d* [3,4]. *Acaryochloris* is the only example of a Chl-*d*-centered photosystem that could be recognized as a specialization in utilizing far-red light for oxygenic photosynthesis [4,5].

*Acaryochloris marina* MBIC11017, which has become the most analyzed type-strain today, was first discovered in 1996 as a symbiont with ascidians in Palau Sea waters as the first cyanobacterium to biosynthesize Chl *d* [3]. Since then, *Acaryochloris* species have been reported to be distributed in various environments, including marine stromatolites, a saline lake epilithic biofilm, and the aquatic plant rhizosphere of freshwater streams [6]. The complete genome of *A. marina* MBIC11017, reported in 2008, consists of a chromosome (6.5 Mb) and nine plasmids, giving it a total genome size of 8.3 Mb, still one of the largest genomes for unicellular cyanobacteria to date [7]. This huge genome and its extensibility could allow *Acaryochloris* to adaptively change into various niches. The fact that many *Acaryochloris* strains have been found in various environments confirms this hypothesis. Gene acquisition through horizontal gene transfer (HGT) acts as the primary driving force for adaptive evolution to a new environmental niche. A set of phycobiliprotein (PBP)-related genes is known as one typical example of HGT in *Acaryochloris* [8]. Among numerous *Acaryochloris* species, only *A. marina* MBIC11017 retains PBP as a photosynthetic antenna complex. PBP-binding bilin pigments as a photosynthetic antenna are distributed among cyanobacteria and red algae. Genes for PBPs appear to have been lost in the common ancestor of *Acaryochloris* and some marine cyanobacteria. However, *A. marina* MBIC11017 is the only *Acaryochloris* strain that retains PBP, which is thought to have reacquired the PBP genes through HGT [8]. Additionally, some *Acaryochloris* strains have been reported to acquire genes related to the nitrogen-fixing enzyme nitrogenase by HGT and are actually capable of nitrogen-fixing growth [6,9]. In both cases, the gene transfer event appears to have occurred in units of gene clusters containing many genes, suggesting that *Acaryochloris* may have a mechanism which allows it to acquire such a large gene cluster and provides an excellent model for investigating the mechanism of HGT of such large gene clusters. However, most *Acaryochloris* genomes reported so far have been sequenced with short-read types of next-generation sequencers, and registered as many contigs, with complete genome sequences limited to only three species.

In this paper, we report the complete genome of *Acaryochloris marina* MBIC10699 (previously registered as *Acaryochloris* sp. MBIC10699), a strain of *Acaryochloris* without PBP isolated from the same ocean as *A. marina* MBIC11017. Since *A**. marina* MBIC10699 was found to be phylogenetically very closely related to *A. marina* MBIC11017, a genomic comparison of these two strains allows us to examine how the *Acaryochloris* lineage leading to *A. marina* MBIC11017 acquired the PBP genes through HGT and what changes in the photosynthetic reaction centers and antenna complexes occurred as a result of the PBP reacquisition. Comparative analysis showed that there is diversity in plasmid composition and plasmid-coding genes in these two *Acaryochloris* strains, and that phenotypic differences, including the presence or absence of PBP, can be explained by differences in plasmid gene composition. It suggests that *Acaryochloris* has acquired various phenotypes through HGT by using multiple giant plasmids as a gene pool.

## 2. Materials and Methods

### 2.1. Isolation and Cultivation of Acaryochloris

*Acaryochloris marina* MBIC10699 was purchased from Biological Resource Center, NITE (NBRC; Kisarazu, Japan), as *Acaryochloris* sp. NBRC102871. Contaminant bacteria, which showed small white spherical colonies on an agar plate in the original cultures were eliminated by repeated single-colony isolation. The isolated pure culture was maintained on agar plates in the laboratory. *Acaryochloris* strains were grown in a 0.5% (*v*/*v*) Daigo IMK medium with 3.6% (*w*/*v*) artificial seawater (Marine Art SF-1) with white fluorescent light exposure at 10 µE/m^2^ s (FRL40SW; Hitachi, Tokyo, Japan) at 26 °C.

### 2.2. Pigment Extraction

Total pigments from *Acaryochloris* cells were extracted according to the previous report [10]. After adding methanol to the collected cells (final 90% (*v*/*v*)), the cells were disrupted by sonication (TOMY UD-201; output: 3, duty: 30, 10 s; TOMY, Tokyo, Japan) and then left on ice for 30 min for pigment extraction. The supernatant collected by centrifugation at 15,000 rpm for 20 min (MX-300, AR015-24; TOMY) was used for HPLC analysis. HPLC analysis (Shimadzu LC20-AD, Shimadzu, Kyoto, Japan) was performed according to Zapata’s method [11] and Chl *a* and *d* were detected by absorption at 440 nm and 690 nm, respectively. Quantification of Chl *a* and Chl *d* was performed by preparing standard curves of peak areas on HPLC chromatograms with standard pigments.

### 2.3. Extraction of Genomic DNA

Cells grown in 45 mL IMK medium were corrected and suspended in 500 µL TE buffer. After three cycles of freezing and thawing, 100 µL of 50 mg/mL lysozyme solution was added to the cell suspension, followed by 1 h incubation at 37 °C. After washing of the cells with TE, cells were suspended in 600 µL of DNA extraction buffer (DNA suisui; Rizo Inc., Tsukuba, Japan) with 3 µL of RNase solution and incubated at 70 °C for 10 min. After cooling it down to room temperature (RT), 600 µL of TE-saturated phenol was added and mixed by inverting the tube up and down, then slowly stirred at RT for 15 min. The upper phase was recovered after centrifugation (15,000 rpm, 5 min, RT) and then an equal volume of PCI solution (phenol/chloroform/isoamyl alcohol, 25:24:1) was added and stirred slowly for 15 min at RT. This wash step with PCI was repeated once, and the upper phase was mixed with an equal volume of CIA (chloroform/isoamyl alcohol, 24:1) and stirred slowly for 5 min at RT. The upper phase was recovered after centrifugation (15,000 rpm, 5 min, RT) and 1/10 volume of 3 M sodium acetate solution and an equal volume of 2-propanol were added, mixed gently, and incubated on ice for 10 min. After centrifugation (15,000 rpm, 10 min, 4 °C), the pellet was washed with 70% ethanol and dried to remove ethanol completely. Genomic DNA was dissolved in 30 µL of TE buffer.

### 2.4. Whole Genome Sequencing of A. marina MBIC10699

The obtained high molecular weight DNA was shipped to the Oral Microbiome Center in Taniguchi Dental Clinic in Japan for long-read and short-read sequencing. Briefly explained, the paired-end (2 × 150-bp) DNA library was prepared using the MGIEasy FS PCR-free DNA library prep set (MGI Tech., Shenzhen, China), according to the manufacturer’s instructions. DNBSEQ-G400 sequencing yielded 1,860,686 paired-end reads. For long-read sequencing, DNA library was prepared using a ligation sequencing kit (SQK-LSK-109; Oxford Nanopore Technologies, Ltd. (ONT), Oxford, UK). An R9.4.1 flow cell (FLO-MIN106) was used to sequence a DNA library using a GridION X5 system (ONT).1- 06). After a 24 h run, the FAST5 format file was base called using Guppy v.3.6.0 (ONT), which generated 56,313 reads.

### 2.5. Quality Control of Sequencing Reads

All raw sequencing data were pre-processed to reduce low-quality bases. For the BGI-seq reads, fastp 0.23.0 [12] pre-processor was used with the “-q 20 -t 1 -T 1 -l 20” option. For the Oxford Nanopore reads, nanofilt 2.7.0 [13] was used with the “-q 10 -l 500 --headcrop 75” option.

### 2.6. Genome Assembly and Gene Annotation

For the genome assembly, pre-processed short reads and long reads were hybrid assembled using Flye 0.2.8 [14]. The output of Flye was polished using Pilon [15] three times, generating a single circular sequence for the chromosome and another four circular plasmid sequences. The chromosome sequence was rotated to the first nucleotide of the 100 bp upstream of the *dnaA* gene. The genome sequence was then annotated using the annotation pipeline DFAST 1.5.0 [16] provided by DDBJ. DFAST automatic annotation predicted 6800 coding sequences and 6 rRNA genes and 72 tRNA genes. The sequencing depth of the chromosome and four plasmid sequences were calculated using BBmap (BBMap-Bushnell B.-sourceforge.net/projects/bbmap/, accessed on 15 March 2021). The complete genome sequence and annotation of *A. marina* MBIC10699 was deposited at DDBJ under accession numbers AP026075 (Chr), AP026076 (pREC1), AP026077 (pREC2), AP026078 (pREC3), and AP026079 (pREC4). Raw sequencing data were deposited in the DDBJ SRA database under BioProject number PRJDB13468 and BioSample number SAMD00467986.

### 2.7. Comparative Genomics

*A. marina* MBIC10699 and *A. marina* MBIC11017 genome sequences were compared using pairwise average nucleotide identity (ANI) and dot plot analysis. For ANI calculation, orthoANI 0.5.0 [17] was used. For dot plot comparison, D-GENIES web server (http://dgenies.toulouse.inra.fr (accessed on 23 February 2022) was used [18]. Annotation of genes by KEGG was performed by GhostKOALA. Homologous genes were searched by blastp (e-value cut off: 1 × 10^−9^), and the sequence alignment was visually confirmed. When a global alignment could not be confirmed, it was eliminated.

### 2.8. Core genome Phylogenic Analysis

To determine the phylogenic relationship between 26 *Acaryochloris* strains with out-group strains, core protein alignments were generated using roary 3.13.0 [19] with the following options: -e –mafft -r -qc -cd 90 -i 90 –group limit 70000. The maximum likelihood tree was constructed using IQ-TREE version 2.0.3 with 100 bootstraps using the best-fit model (LG + F + R10) determined in ModelFinder.

## 3. Results and Discussion

### 3.1. Comparative Analysis of Pigment Contents of Two Acaryochloris Strains

*Acaryochloris* sp. MBIC10699 (hereafter MB10699) differs significantly in cell coloration compared with the type-strain *A. marina* MBIC11017 (MB11017). Comparison of their cellular spectra showed a clear difference in the 550–650 nm absorption peak derived from phycobiliprotein (PBP), confirming the presence and absence of PBP in MB11017 and MB10699, respectively (Figure 1). HPLC analysis of methanol cell extracts showed that the two *Acaryochloris* strains had the same chlorophyll (Chl) and carotenoid compositions. The amount of cellular Chl *d* in MB10699 was about 1.5 times higher than that in MB11017, as reflected in the Chl *a/d* ratios: 0.044 in MB10699 and 0.052 in MB11017 (Figure 1). The major carotenoids common to MB10699 and MB11017 are thought to be zeaxanthin and α-carotene [20]. These results indicate that the two strains are nearly identical in photosynthetic pigment composition except for the presence or absence of PBP, even though there is a slight difference in Chl *d* content. MB10699, which does not have PBP as an antenna protein, is thought to employ another antenna complex, Chl-bound Pcb (prochlorophyte chlorophyll-binding protein), as the primary antenna [21,22]. Therefore, it is expected that MB10699 contains a larger amount of Pcb antennae than MB11017 to accommodate higher amounts of Chl *d*.

### 3.2. Overall View of the Complete Genome of A. marina MBIC10699

The complete genomic structure of MB10699 was determined by combining Oxford Nanopore long-read and MGI-seq short-read sequencing. The genome of MB10699 consists of one circular chromosome (Chr; 6.4 Mb) and four giant plasmids (pREC1, 393 kb; pREC2, 329 kb; pREC3, 303 kb; and pREC4, 205 kb) with a total genome size of 7.6 Mb (Figure 2), which is significantly less than that of MB11017 (8.3 Mb). The short-read coverage depths were almost the same for Chr and the four plasmids, suggesting that these plasmids exist in the same number of copies as Chr (data not shown). The GC content and coding capacity were 47.0% and 82.3%, respectively, which were almost identical to those of MB11017, with 47.0% and 84.1%, respectively. All 6 rRNAs and 72 tRNAs were encoded by Chr, and 3 CRISPRs were encoded by Chr (1) and pREC1 (2). The number of CDSs in the MB10699 genome was 6813, about 80% of the 8528 CDSs found in MB11017 (Table 1). Pairwise average nucleotide identity (ANI) calculations showed that MB10699 had a value of 98.0 relative to MB11017, indicating that these two *Acaryochloris* strains are closely related enough to be categorized as the same species [23,24]. A phylogenetic tree based on the 897 number of concatenated protein sequences among 26 *Acaryochloris* species with *Cyanothece* sp. PCC7425 and *A. thomasi* RCC1774 showed that *Acaryochloris* is divided into three major clades, and MB10699 and MB11017 are the most closely related (Figure 3, Appendix A). Based on the results of ANI and this phylogenetic analysis, MB10699 was renamed *A. marina* MBIC10699 as the same species as *A. marina* MBIC11017 in the taxonomy, and the genome information was registered as *A. marina* MBIC10699. Since MB11017 is the sole strain that retains PBP among the 26 *Acaryochloris* strains, including MB10699, genome comparison between MB10699 and MB11017 provides an excellent example to trace the evolution of the reacquisition of PBP genes through HGT.

Dot plot analysis of MB10699 and MB11017 genomes revealed global sequence identity between their Chrs (Figure 4). In contrast, there was low homology and large divergence among plasmid sequences. In support of this, an examination of the genes annotated by KEGG in these two *Acaryochloris* strains showed that the composition of the genes encoded by Chr was almost identical, with MB10699 and MB11017 specific to each with only 0.6% (15) and 0.75% (12), respectively (Figure 5, Appendix A). In contrast, for plasmid-encoded genes, 25% (63) of the genes in MB10699 and 43% (146) of the genes in MB11017 were unique, with no other homologous genes in the other (Appendix A). There are four plasmids in MB10699 (pREC1-4) and nine in MB11017 (pREB1-9). Focusing on the homology of plasmids between the two strains, significant homology was observed only between pREC1 (MB10699) and pREB2 (MB11017) (Figure 4). In fact, characteristic metabolic genes such as genes for uridine kinase and aliphatic amidase are conserved in these plasmids, suggesting that these two plasmids have derived from the same origin. Additionally, a region of pREC2 showed significant homology to the sequence within pREB6. Although these plasmids may share the same origin, their size was quite different, 329 kb (pREC2) and 172 kb (pREB6), suggesting that many more genes have been acquired in pREC2 than in pREB6. No other combinations showed significant homology among the other plasmids.

It has been reported that plasmids contain mobilization factors such as *oriT* and relaxases as factors involved in their mobility from cell to other cells via conjugation [25,26]. Among these factors, it has been proposed that plasmids can be classified based on the sequences of relaxases, which are essential for mobility. The relaxase gene was found one by one on all four pREC plasmids in MB10699, and on all pREB plasmids except pREB9 in MB11017. Since the relaxase gene is a unique factor for each plasmid, the correspondence of the plasmids between the two *Acaryochloris* strains was analyzed based on the homology of this relaxase gene, *mobF* (Figure 6). Phylogenetic analysis using the amino acid sequences of MobF revealed that they were classified into two major clades, each of which further branched into two subgroups. The phylogenetic relationship of relaxases suggested that pREC1 is of the same origin as pREB2, pREC2 as pREB6, pREC3 as pREB4,5, and pREC4 as pREB7,8, respectively. In fact, dot plot analysis and ANI comparisons indicated high similarities in the pairs of pREC1/pREB2 and pREC2/pREB6; thus, this plasmid classification based on phylogenetic analysis of relaxases is considered reliable. From these results, the pREC1/pREB2 pair and the pREC2/pREB6 pair are likely derived from common ancestral plasmids, respectively, and similarly the pair of pREC3/pREB4(5) and pREC4/pREB7(8) are also assumed to be derived from the same origins. There is no MB10699 plasmid in the clade corresponding to pREB1 and pREB3, suggesting that these plasmids were either acquired after the divergence of the two species or lost at MB10699 after the divergence. This suggests that HGT of PBP-related genes in pREB3 possibly occurred in plasmid units.

The high degree of conservation of Chr sequences in these two strains and the high diversity among the plasmids suggest that various mutations in the plasmids have played a major role in the divergence of these two strains. Some plasmids that appear to be derived from the same origin show high homology while others do not, suggesting that the loss and gain of genes on the plasmids have not occurred evenly and that mutation frequency is different among the plasmids. Indeed, the presence or absence of PBP, the most characteristic phenotypic difference between MB10699 and MB11017, is determined by a group of genes accumulated on pREB3. Thus, even among closely related *Acaryochloris* strains, there is a high variety of gene contents and high flexibility of genetic variation on the plasmids, suggesting that *Acaryochloris* has taken advantage of this plasmid gene extensibility to adapt to various environments.

### 3.3. Comparative Analysis of Genes Conserved in Plasmids between Two Acaryochloris Species

To look at plasmid divergence in detail, we annotated the genes encoded in the plasmids of the two *Acaryochloris* strains with KEGG and selected genes unique for the respective species (Appendix A). A total of 63 specific genes were found in the MB10699 plasmids, 27 of which were not present in MB11017, including its Chr. Among the 63 genes, genes involved in sulfur metabolism (phosphoadenosine phosphosulfate reductase, sulfate adenyltransferase, and sulfur dioxygenase), metal ion transporters, and glycolysis (α-amylase, α-glucosidase, glycoside/pentoside: cation symporter) were found. In contrast, 146 genes unique to MB11017 plasmids were found, of which 60 were unique to MB11017 that were not present in MB10699 including its Chr. Genes specific to MB11017 plasmids include PBP-related genes, another set of ATP synthase, and bidirectional hydrogenase-related genes, as well as pyruvate-ferredoxin/flavodoxin oxidoreductase, acetyl-CoA synthetase, NAD(P)H-quinone oxidoreductase, fatty acid CoA ligase, and other carbon metabolism genes. We also found genes involved in the biosynthesis of porphyrins, such as divinyl chlorophyllide reductase and *hemL* (GSA) in MB11017 plasmids. To determine how these plasmid-specific genes were acquired, homologous regions were aligned for pREC1 and pREB2, considered having a common plasmid origin (Appendix A). The ORFs characteristic of pREC1 were found to form several clusters on the plasmid, with transposase genes located in the vicinity. This suggests that gene gain or loss on plasmids occur in clusters containing multiple genes. Alternatively, there may be ‘hot-spot’ loci where gene gain or loss is prone to occur by transposases, and gene gain or loss events may have been concentrated around these positions.

### 3.4. PBP-Related Genes

In MB11017, *cpcA/B* (α and β subunits of phycocyanin), *cpcC/D/G* (linker polypeptides that connect discs composed of phycocyanin heterodimers), *hemH* (ferrochelatase), *ho* (heme oxygenase), *pcyA* (phycocyanobilin oxidoreductase in the biosynthesis of phycocyanobilin), and *cpcE/F* (phycocyanobilin lyases that bind phycocyanobilin to the CpcA/B apo-protein) are encoded in the plasmid pREB3. Four paralogs of *cpcA/B*, three paralogs of *cpcG*, and two each of *cpcC/D*, *hemH*, and *ho* form multiple operons within pREB3. Cellular spectra of MB10699 indicated that it does not have PBPs as an antenna complex (Figure 1). As mentioned above, MB10699 does not harbor a plasmid corresponding to pREB3, and many of the PBP-related genes were not found in the entire genome. In MB11017, *hemH*, *ho*, and *pcyA*, which are involved in phycocyanobilin biosynthesis, are present as four, three, and two paralogous genes, respectively, and are encoded in both Chr and pREB3 (Figure 7). Although no paralogs of these three genes were found in the plasmids of MB10699, we found an equal number of paralogs of each gene in the Chr as in MB11017 Chr. Phycocyanobilin functions as a chromophore of the photoreceptor cyanobacteriochromes, in addition to PBPs [27,28]. Indeed, two paralogs of phycocyanobilin:ferredoxin oxidoreductases (PcyA) encoded by Chr and pREB3 show different biochemical properties, suggesting that these PcyAs are specialized in providing chromophores to apo-proteins of cyanobacteriochrome and phycocyanin, respectively [29]. PcyA conserved in Chr of MB10699 shows an identical amino acid sequence to that conserved in Chr of MB11017, suggesting that Chr-encoding PcyA is specialized to provide chromophores to cyanobacteriochrome. In MB10699, only *apcB*, which encodes β-subunit of allophycocyanin, is conserved in Chr, and this gene is similarly conserved in Chr in MB11017, while it has not yet been examined whether this *apcB* is actually expressed in MB11017. Since *apcB* is conserved even in MB10699, this *apcB* may serve some different functions rather than the allophycocyanin subunit of PBPs.

### 3.5. Genes Related to Alternative ATP Synthase and Hydrogenase

In MB11017, eight genes for *atpABCDEFGH*, subunits of F-type ATP synthase, are present in Chr, and these paralogs of the seven genes except *atpH* are also conserved in pREB4 (Figure 7). It is not known whether these two sets of ATP synthase share the Chr-coded δ subunit (AtpH) or whether the pREB4-encoded ATP synthase functions in a form that has lost its δ subunit (N-type ATP synthase). When functioning as an N-type ATP synthase, it is thought to transport Na^+^ instead of H^+^. Other marine cyanobacteria such as *Cyanothece* also conserved two sets of ATP synthase genes lacking one δ subunit, suggesting that this is a conserved form of the gene with some significance. In addition to ATP synthase, *hoxEFUYH* genes, which encode hydrogenase subunits, and *hypABCDEF* genes, which are involved in hydrogenase maturation, are conserved in pREB4 of MB11017 (only *hypAB* has their paralogs in Chr) (Figure 7). *hoxEFUYH* genes encode a bidirectional hydrogenase, which catalyzes the conversion of protons and molecular hydrogen in both directions. These genes for plasmid-encoded ATP synthase (alternative ATP synthase) and *hox*-hydrogenase are not conserved in MB10699. A total of 26 *Acaryochloris* strains reported to date were tested for the presence of the alternative ATP synthase and *hox*-hydrogenase genes (Figure 3). Since these gene sets were also found in several other *Acaryochloris* strains, no correlation was found between the acquisition of PBP and the conservation of this alternative ATP synthase and *hox*-hydrogenase. Interestingly, the conservation of both gene sets for the alternative ATP synthases and *hox*-hydrogenase was perfectly consistent in *Acaryochloris* species without any exceptions. Given the high correlation between the alternative ATP synthase and *hox*-hydrogenase conservation, the gain or loss of these genes is expected to occur synchronously. The conservation of these genes is inconsistent with the phylogenetic relationship of *Acaryochloris* (Figure 3), and species that conserved the alternative ATP synthase and *hox*-hydrogenase genes appear mosaic in all clades. This suggests that both gene sets for the alternative ATP synthase and *hox*-hydrogenase were retained at origin before these *Acaryochloris* species diverged, but that some species lost both genes after divergence. The complete coexistence of both of these genes suggests that they are functionally related. It has been suggested that bidirectional hydrogenase can release excess reducing power in fermentation by proton reduction [30]. Then, the energy stored as molecular hydrogen is extracted by the reverse reaction to produce reducing power (NADPH) when needed, and the resulting proton gradient drives ATP synthase to generate ATP. Since this combination of bidirectional hydrogenase and ATP synthase enables the generation of NADPH and ATP using molecular hydrogen, it may play an important role in conditions where both photosynthesis and respiratory electron transfer systems do not fully function, such as dark and anaerobic conditions.

### 3.6. Genes Related to Photosynthetic Reaction Centers, Electron Transfer, and Chl-Binding Antenna Complex

Comparative analysis was performed between MB10699 and MB11017 for genes involved in photosystems, antenna complexes, and photosynthetic electron transfer systems that are potentially affected by the presence or absence of PBP (Figure 7). All genes encoding subunits of PSI, PSII, and the cytochrome *b_6_f* complex in MB11017 were also conserved in MB10699 and nearly all of their amino acid sequences were identical in these two *Acaryochloris* strains, although some of them were found with a different number of paralogs. While three *psbA* paralogs (AM1_0488, 2166, 2889) are found in MB11017, four *psbA* paralogs (AM10699_32560, 54700, 47980, 47970) are found in MB10699, of which AM10699_32560 and AM1_0488 are identical, and the remaining three *psbA* paralogs were identical and perfectly matched with AM1_2166/2889. Four paralogous genes of *psbU* gene are present in MB11017, two of which are conserved in its Chr and the other two in pREB4 and pREB7. However, in MB10699, only two *psbUs* (AM10699_07930, 17380) were found in Chr and no *psbU* paralogs in plasmids. Although PsbU plays a role in stabilizing energy transfer between PSII and PBPs [31], *psbU* gene was also conserved in MB10699. The amino acid sequences of two plasmid-encoded PsbUs (AM1_D0138 and G0114) are significantly different from those of the Chr-encoded PsbUs (Appendix A), suggesting that these plasmid-encoded PsbUs are involved in the complex formation of PBP and PSII in MB11017. In MB10699, seven *petF* paralogs encoding ferredoxin are conserved in its Chr, and in MB10117, another *petF* paralog is found in the plasmid in addition to the seven *petF* paralogs in its Chr. The *petF* lost in MB10699 is conserved in pREB3 of MB11017, which is most likely the ferredoxin involved in PBP biosynthesis (PcyA). *Acaryochloris* retains a unique type of antenna complex Pcb that binds Chl *a/d*, encoded by *isiA* that shows significant homology to *psbC* (CP43) [21]. MB10699 has seven *isiA* paralogs, and eight in MB11017. One extra paralog in MB11017, AM1_C0105, is also encoded by pREB3.

Most of these photosynthesis-related genes were identical in the amino acid sequence between the two *Acaryochloris* strains; however, significant differences were found in the following genes. *petE*, which encodes plastocyanin, showed 95% identity between the two strains, and other genes such as *petF* (AM1_5442) and FNR (*petH*) (AM1_2942) had 99% identity with their counterparts in MB10699. Three *isiA* genes, AM1_3362, 3364, and 3366 of MB11017 are consecutively located on Chr and form an operon, and the corresponding, AM10699_02480, 02490, and 02500 of MB10699 form a similar operon on Chr. AM10699_02480 and the corresponding paralog AM1_3362 showed 88% identity, which is significantly lower than other paralog pairs. AM10699_02490 and 02450, corresponding to AM1_3364 and 3366 in the same operon, showed 99% identity with significant differences. MB10699 should employ Pcbs (IsiA) as the main photosynthetic antenna complex because of the absence of PBPs. Compared to MB11017, the number of *isiA* paralogs was rather reduced by one in MB10699, and there was significantly less sequence homology in the operon with three consecutive *isiA*s between the two species. These three *isiAs* may represent examples of unique evolution influenced by the presence or absence of PBP. Further studies are required to determine how the amino acid differences in these IsiAs have altered their function.

### 3.7. Availability of Carbon Sources in Two Acaryochloris Strains

Recently, MB10699 and MB11017 were reported to differ in their ability to use exogenous carbon sources [10]. For example, MB10699 was capable of mixotrophic growth with sucrose and glucose as carbon sources, whereas MB11017 was lethal in cultures with these carbon sources. Conversely, MB11017 tolerates glycerol as a carbon source, but MB10699 was lethal in cultures containing glycerol. We discuss these phenotypic differences through a comparison of their genomic information. Comparison of sucrose metabolism genes in these two *Acaryochloris* strains showed that there are equal numbers of paralogs for sucrose phosphate, sucrose synthase, invertase, amylosucrase, and α-glucosidase genes encoded by Chr, but another paralog of α-glucosidase is conserved in pREC2 of MB10699 (AM10699_61640). Furthermore, α-amylase and glycoside/pentoside: cation symporter, are also specifically conserved in the plasmids of MB10699, suggesting that these plasmid-encoded genes are responsible for the superior sucrose/glucose metabolism in MB10699 compared to MB11017. The effect of glycerol addition on cyanobacteria was reported in *Synechococcus* sp. PCC7002, where glycerol had a negative effect on growth in mutants with loss of plastoquinone pool regulation [32]. In MB11017, the genes for bidirectional hydrogenase and NAD(P)H-quinone oxidoreductase are specifically conserved in the plasmids, and these genes seem to be involved in the redox regulation of plastoquinone pool. Thus, the different genetic composition of their plasmids would be expected to contribute to phenotypic differences in the availability of carbon sources, supporting that the differences in plasmid-constitutive genes determine the phenotypic differences between these two *Acaryochloris* strains. It is necessary to consider the effects of these plasmid-coded genes in combination with transcriptome analysis in the future.

## 4. Conclusions

In this report, we sequenced the complete genome of *Acaryochloris marina* MBIC10699, a PBP-less *Acaryochloris*, and found that it is closely related to type-strain *A. marina* MBIC11017. Chr sequence homology between these two *Acaryochloris* strains was extremely high, with no major inversions or duplications, and their genetic composition was largely identical. However, the plasmid composition and the genes encoded in the plasmids were significantly different, and the presence or absence of PBP was also explained by the genes encoded in the plasmids. Although amino acid sequences of most photosynthesis-related genes were identical between the two *Acaryochloris* strains, we found genes with significantly lower homology and genes that differed in the number of their paralogs, suggesting that these differences resulted from the presence or absence of PBPs. Furthermore, the differences in available carbon sources in the two *Acaryochloris* strains also appear to result from differences in the genes encoded in their plasmids, suggesting that most of the phenotypic differences of the two *Acaryochloris* strains are due to differences in the genes constitutive in the plasmids. Comparison of genomic information between these two closely related *Acaryochloris* strains reveals that *Acaryochloris* has acquired and altered traits through gene acquisition using a highly extensible plasmid. This finding provides the basis for a detailed analysis of how horizontal gene transfer of large gene clusters occurs in *Acaryochloris*.

## Figures and Tables

**Figure 1 microorganisms-10-01374-f001:**
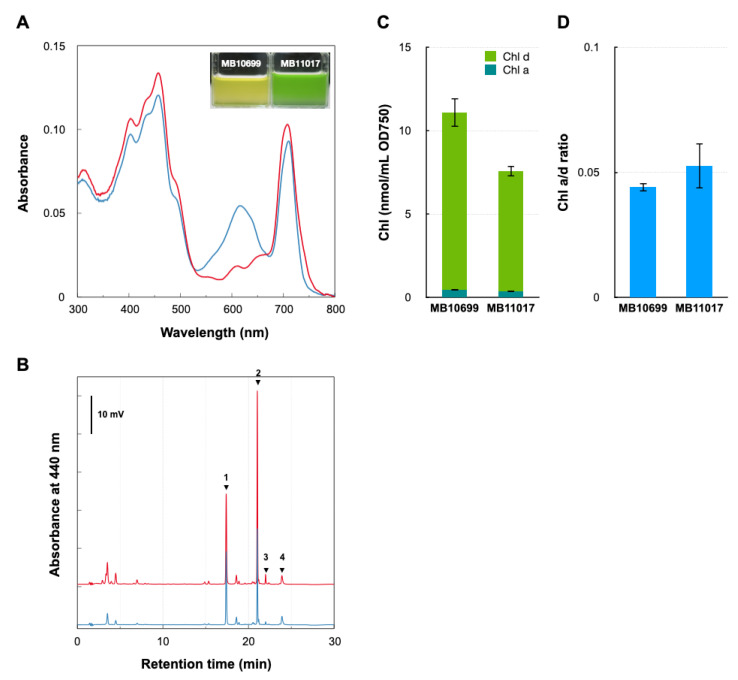
Comparison of cellular pigments between two *Acaryochloris* strains. (**A**) In vivo cellular spectra of *A*. *marina* MBIC10699 (red) and *A. marina* MBIC11017 (blue). Inset: picture of each culture. (**B**) HPLC profile of total pigments extracted with methanol from cells of *A*. *marina* MBIC10699 (red) and *A. marina* MBIC11017 (blue) monitored by absorption at 440 nm. Each arrow indicates zeaxanthin (1), Chl *d* (2), Chl *a* (3), and α-carotene (4), respectively. (**C**) The amounts of Chl *a* (dark green) and Chl *d* (green) were measured in two *Acaryochloris* strains. (**D**) The ratio of cellular Chl *a/d* amounts in the two *Acaryochloris* strains.

**Figure 2 microorganisms-10-01374-f002:**
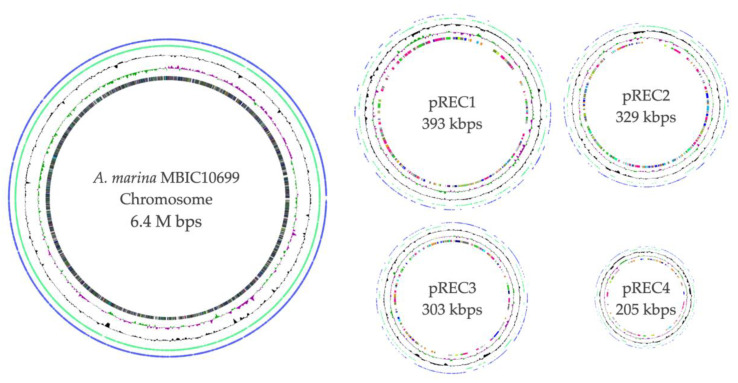
Genome structure of *A*. *marina* MBIC10699. The different rings represent, from outside to inside, all CDS (counterclockwise: blue, clockwise: green), deviation from the average GC content, GC skew, and CDS with colors corresponding to COG categories. For visualization, all plasmids are represented on a 10-fold scale relative to the chromosome.

**Figure 3 microorganisms-10-01374-f003:**
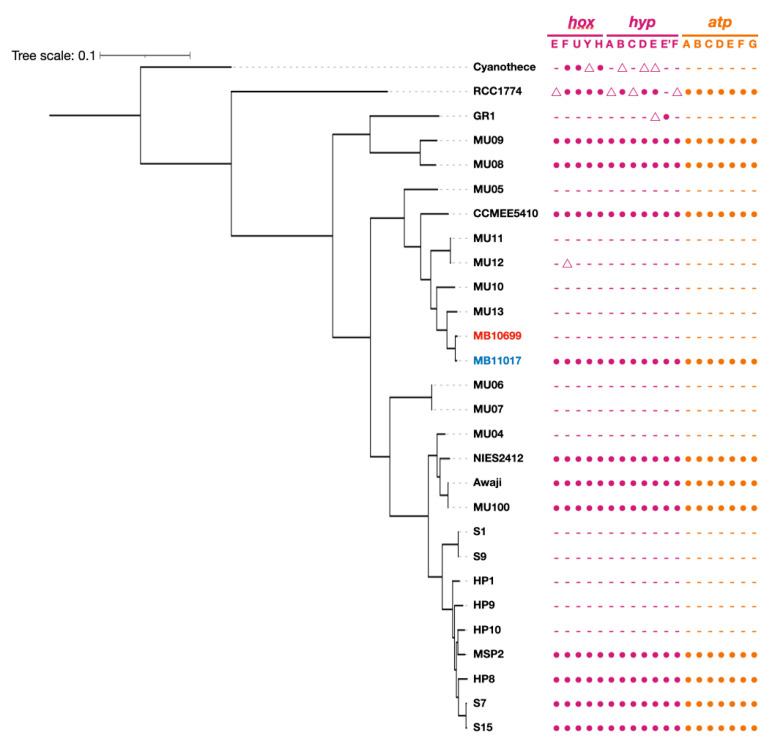
Phylogenic tree of *Acaryochloris* species. A maximum likelihood phylogenic tree of *Acaryochloris* strains was constructed from genome-wide concatenated protein sequences from single copy orthologous genes (*n* = 897). The tree was outgroup-rooted with *Cyanothece* sp. PCC7425 (Cyanothece). Another sister group of *Acaryochloris* strains, *Acaryochloris thomasi* RCC1774 (RCC1774), which biosynthesizes Chl *b* instead of Chl *d*, was used along with previously reported *Acaryochloris* strains. *A. marina* MBIC10699 and MBIC11017 are highlighted in red and blue, respectively. The symbols indicate the presence or absence of genes with significant homology to *hoxEFUYH*/*hypABCDEE’F* (purple) and *atpABCDEFG* (orange) genes encoded in pREB4 of MB11017 in each strain. Circles indicate the conservation of genes with significant homology (e-value < 1 × 10^−100^), and triangles indicate genes that show some degree of homology with e-values between 1 × 10^−100^ and 1 × 10^−50^.

**Figure 4 microorganisms-10-01374-f004:**
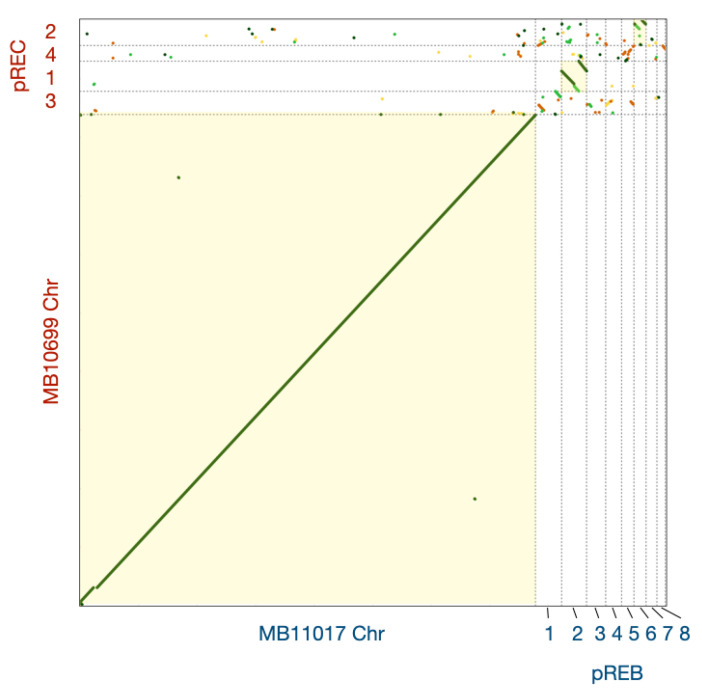
Dot plot analysis between *A*. *marina* MBIC10699 and MBIC11017. Dot plot alignment of *A**. marina* MBIC10699 (vertical axis) versus *A. marina* MBIC11017 (horizontal axis). The color of each dot indicates the degree of identity, with 75% or more represented by green, 75–50% by light green, 50–25% by brown, and 25% or less by yellow, respectively. Matrices of combinations in which significant homology was detected are highlighted in light yellow.

**Figure 5 microorganisms-10-01374-f005:**
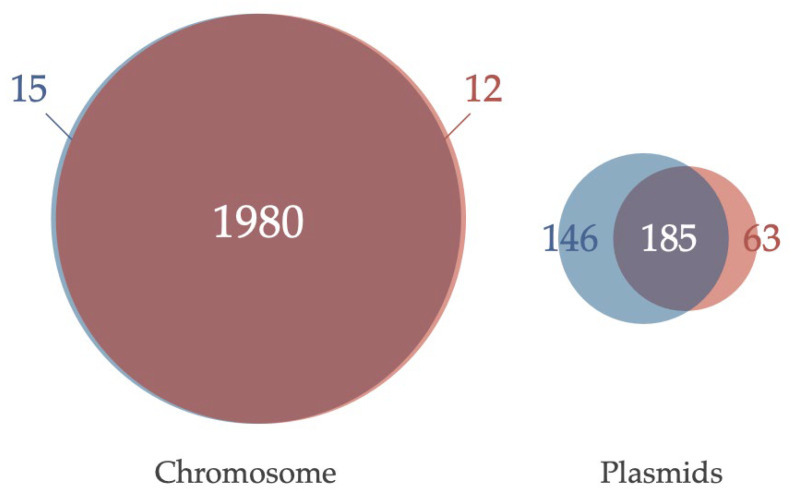
Common and unique genes in the two *Acaryochloris* strains. Euler diagram showing the common genes between *A**. marina* MBIC10699 (red) and *A. marina* MBIC11017 (blue) for each chromosome and plasmids, respectively. Genes annotated by KEGG with the same ID were considered common. The numbers indicate the number of genes in the fraction.

**Figure 6 microorganisms-10-01374-f006:**
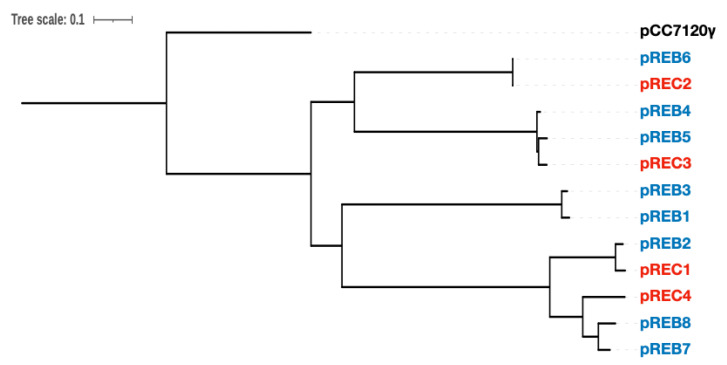
Phylogenetic relationship of each plasmid based on relaxase gene homology. The phylogenetic relationship of each plasmid was classified on the basis of the amino acid sequence of the relaxase (MobF) protein encoded in each plasmid. The plasmids shown in red are from *A. marina* MBIC10699, and the plasmids shown in blue are from *A. marina* MBIC11017. MobF encoded by pCC7120γ (black), the plasmid from *Anabaena* sp. strain PCC7120, was used as the outgroup.

**Figure 7 microorganisms-10-01374-f007:**
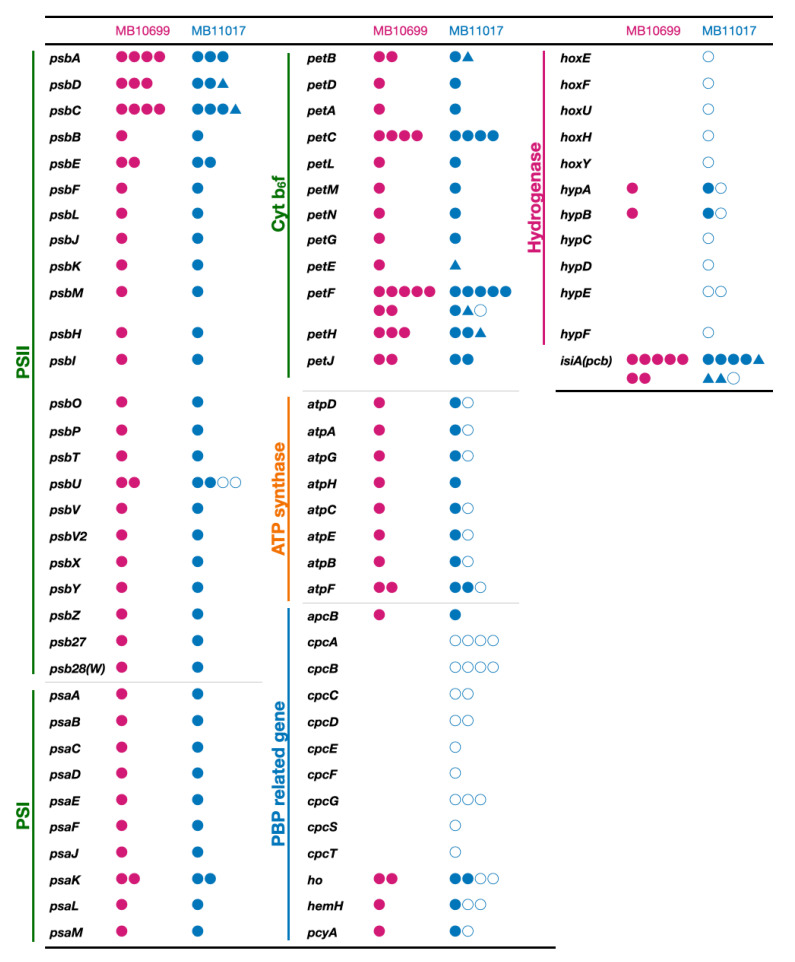
Comparison of the number of gene paralogs related to photosynthesis and electron transfer systems. The number of paralogs of genes related to photosystem I (*psa*), photosystem II (*psb*), cytochrome *b_6_f* and electron transfer (*pet*), ATP synthase (*atp*), PBP, and hydrogenase are indicated by the number of red (*A. marina* MBIC10699) and blue (*A. marina* MBIC11017) symbols. Circles indicate completely identical amino acid sequences between the two strains and triangles indicate that the homology is not 100% (shown in *A. marina* MBIC11017). Filled circles indicate genes encoded by chromosome and empty circles indicate plasmid-encoded genes.

**Table 1 microorganisms-10-01374-t001:** General features of the genome of *A*. *marina* MBIC10699.

	Chr	pREC1	pREC2	pREC3	pREC4	Total
Total Sequence Length (bp):	6,415,507	393,608	329,949	303,490	205,174	7,647,728
Number of Sequences:	1	1	1	1	1	5
Gap Ratio (%):	0.0	0.0	0.0	0.0	0.0	0.0
GC content (%):	47.3	45.1	46.7	46.1	43.2	47.0
Number of CDSs:	5674	366	283	280	193	6813
Coding Ratio (%):	83.4	76.0	75.3	78.1	68.7	82.3
Number of rRNAs:	6	0	0	0	0	6
Number of tRNAs:	72	0	0	0	0	72
Number of CRISPRs:	1	2	0	0	0	3

## Data Availability

The complete genome sequence and annotation of *A. marina* MBIC10699 were deposited at DDBJ under accession numbers AP026075 (Chr), AP026076 (pREC1), AP026077 (pREC2), AP026078 (pREC3), and AP026079 (pREC4). Raw sequencing data were deposited in the DDBJ SRA database under BioProject number PRJDB13468 and BioSample number SAMD00467986.

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
