# Peer review of "Comparative Genomic Analysis of the Marine Cyanobacterium Acaryochloris marina MBIC10699 Reveals the Impact of Phycobiliprotein Reacquisition and the Diversity of Acaryochloris Plasmids"

_microorganisms, 2022, doi:10.3390/microorganisms10071374_

Round 1
Reviewer 1 Report
In this manuscript Yamamoto et al. reported a the comparison of Acaryochloris sp. MBIC10699 with A. marina MBIC11017 in terms of their genome, discussing which genes were reacquired and what changes occurred during evolution. My only comment is a generic comment: the authors made an in-depth genomic analysis between the two compared species. However, taking as example what they said in lines 384-387, saying that a gene is missing or not in one organism compared to the other is not equal to consider the transcriptome and much better the proteome when the two organism are set in the same condition. Is it possible to enrich the analysis made in this manuscript by also considering the gene expression to see what actually the two organisms do?
Author Response
Thank you for spending your valuable time to peer review our paper. I believe our paper got improved based on the points suggested by you.
As you pointed out, in order to understand how the actual metabolic pathway works, it is necessary to complement the analysis with transcriptome and proteome. The sentence that inferred expression levels from the number of gene paralogs was deleted, and a statement was added in the text stating that analysis by transcriptome will be necessary in the future.
Reviewer 2 Report
In this manuscript, Yamamoto and coworkers present a comparative genomic analysis of two cyanobacterium Acaryocloris species, one of which is unique in its ability to produce phycobilins. Their analysis indicates that the metabolic differences observed between the two species are the result of plasmid encoded genes. Thus, providing another example of how horizontal transfer impacts microbial diversity. The manuscript is very well written and the figures are clear and informative. A few additions to the manuscript could help strengthen its impact:
1) Lines 65-67, 270-273: It is suggested that Acaryochloris may have a mechanism to acquire large gene clusters and that associated plasmids may possess hot spots for gene gain and loss. Are there any hints found in the chromosome sequence or plasmids such as Insertion Sequences or transposons that would support these hypotheses?
2) Starting at Line 167 throughout the rest of the manuscript: italics need to be used for genus, species, and gene abbreviations.
3) Lines 198 and 417: Is there a list or reference for the 897 genes included in the phylogenetic tree analysis? This information could be included in a supplemental table.
4) Lines 206-209 and Figure 5: Is there anything interesting regarding the identity of the few chromosomal genes that are specific to each species?
5) Line 219: Change “gene” to “genes”.
6) Line 221: Define “MOB”.
7) Lines 232-241 and 254-273: Are there any hints within the plasmid sequences to determine the origin of these plasmids and associated unique genes or are there other microbes outside of Acaryochloris that possess similar plasmids and genes?
8) Lines 316-328 and Figure 3: Are the alternative ATP synthase and hydrogenase genes chromosomally- or plasmid-encoded (or a mixture of both)?
9) Lines 348-377 and Table 2: Do the plasmid encoded paralogs possess ~100% homology with the chromosomal equivalents or is there sequence divergence?
10) Supplemental Tables 1 and 2: The table legends suggest that genes in black font can be found on the chromosome of the other species. This is likely the case, but it would help to explicitly state if true.
Author Response
Thank you for spending your valuable time to peer review our paper. I believe our paper got improved based on the points suggested by you.
1) Lines 65-67, 270-273: It is suggested that Acaryochloris may have a mechanism to acquire large gene clusters and that associated plasmids may possess hot spots for gene gain and loss. Are there any hints found in the chromosome sequence or plasmids such as Insertion Sequences or transposons that would support these hypotheses?
>In the figure of the alignment of pREC1 and pREB2 (Fig. S1), the loci of the transposon genes were added. We also added the following statement describing the relationship between gene insertions and transposons.
The ORFs characteristic of pREC1 were found to form several clusters on the plasmid, with transposase genes located in the vicinity. This suggests that gene gain or loss on plasmids occur in clusters containing multiple genes. Alternatively, there may be ‘hot-spot’ loci where gene gain or loss is prone to occur by transposases, and gene gain or loss events may have been concentrated around these positions.
2) Starting at Line 167 throughout the rest of the manuscript: italics need to be used for genus, species, and gene abbreviations.
>It was a mistake that happened when copy-pasting the text. We have checked and corrected the mistakes throughout the text.
3) Lines 198 and 417: Is there a list or reference for the 897 genes included in the phylogenetic tree analysis? This information could be included in a supplemental table.
>We added the list of genes for the concatenated sequences as Table S1.
4) Lines 206-209 and Figure 5: Is there anything interesting regarding the identity of the few chromosomal genes that are specific to each species?
>We have added a list of genes specific to each Chrs as Table S2 and S3. As far as I understand, I do not see any unique genes that might affect the trait.
5) Line 219: Change “gene” to “genes”.
>We have revised as you mentioned.
6) Line 221: Define “MOB”.
> We cheged that sentence as follow.
It has been reported that plasmids contain mobilization factors such as oriT and relaxases as factors involved in their mobility from cell to other cells via conjugation
7) Lines 232-241 and 254-273: Are there any hints within the plasmid sequences to determine the origin of these plasmids and associated unique genes or are there other microbes outside of Acaryochloris that possess similar plasmids and genes?
>An comprehensive homology search of the genes encoded in the Acaryochloris plasmid revealed that most of the genes were found in cyanobacterial genomes. Some genes were found to be derived from proteobacteria, but the proportion is not large. Among the cyanobacteria, a large number of genes were found in Leptolyngbya, Cyanothece, and Synechococcus, except for Acaryochloris, so it is thought that many genes were acquired from these organisms. However, this information did not allow us to determine the origin of the plasmids.
8) Lines 316-328 and Figure 3: Are the alternative ATP synthase and hydrogenase genes chromosomally- or plasmid-encoded (or a mixture of both)?
>Fig. 3 shows that the altanative ATP synthase and hydrogenase genes are all plasmid-coded in A. marina MBIC11017. In other Acaryochloris, we confirmed that all these genes are plasmid-coded in the S15 strain where complete genomes are avalable, but in other Acaryochloris, the exact loci are not known because they are reported as draft genomes.
9) Lines 348-377 and Table 2: Do the plasmid encoded paralogs possess ~100% homology with the chromosomal equivalents or is there sequence divergence?
>The PsbU paralogs encoded in plasmids are not identical to the paralogs encoded in Chr, and there are significant differences in the sequence. A figure describing the alignment of the PsbU paralogs was created as Fig S2.
10) Supplemental Tables 1 and 2: The table legends suggest that genes in black font can be found on the chromosome of the other species. This is likely the case, but it would help to explicitly state if true.
>We have chenged the legends as you pointed out.
Genes encoded in the plasmids of A. marina MBIC11017 that are not present in the plasmids of A. marina MBIC10699 were extracted. Genes in blue indicate specific genes that are not present in A. marina MBIC10699 at all, while genes in black indicate that they are present in the chromosome of A. marina MBIC10699.
Reviewer 3 Report
The article by Yamamoto et al. is devoted to the sequencing and subsequent analysis of a strain of Acaryochloris marinus that accumulates chlorophyll d, while being devoid of phycobilis. This phenomenon makes this strain an amazing naturally occurring analogue of the experimentally valuable "mutant/wild-type" systems, which are universally used to study photosynthesis and physiology of these photosynthetic microorganisms.
The article opens up many prospects for further research: is this strain transformable? what will happen if we add phycobilisome to it? what differences in the rates of basic physiological reactions of photosynthesis and respiration distinguish this strain? etc.
In my opinion, the article is performed at a high experimental level and carries a lot of useful information for specialists. Although I do not judge about the level of its writing with respect to English, I am sure that it should be published in the journal MDPI.
Author Response
Thank you for spending your valuable time to peer review our paper. I believe our paper got improved based on the points suggested by you.
The article opens up many prospects for further research: is this strain transformable? what will happen if we add phycobilisome to it? what differences in the rates of basic physiological reactions of photosynthesis and respiration distinguish this strain? etc.
>The technical basis for transformation has been improved in the type-strain, Acaryochloris marina MBIC11017 and random mutations using transposons and heterologous gene expression by shuttle vectors have been reported. There is no report on whether A. marina MBIC10699 can be transformed or not. If it is transformable, it might allow for artificial reacquisition of PBP by introducing a phycobilisome-related genes. Although photosynthetic and respiratory activities have not been measured, it has been reported that the wavelength range available for photosynthesis differs depending on the presence or absence of PBP, and Acaryochloris without PBP does not grow photosynthetically in orange light.
Reviewer 4 Report
This is a good comparative genomic work for Acaryochloris, showing how Acaryochloris has acquired and altered traits through gene acquisition using a highly extensible plasmid. The manuscript is very well written (although some care should be given in the abbrevations, which are many and the Figure legends, which some times are not self-explanatory as they should be), the genomic analysis is solid and the illustration of the results is adequate. I only have some minor comments listed below:
Α. sp. : It not acceptable to abbreviate the genus when it followed by sp. Please correct it as Acaryochloris sp. throught the manuscript, for clarity.
lines 85-91: please explain the NBRC abbreviation; please give details on the isolation area and the nature of the environment it was isolated from (if details are available); please give the wavelength of FRL40SW
lines 99-100: please specify which chl absorbs in which wavelength, for clarity
line 168 and elsewhere: please make sure italics are used for taxa, chl, and genes
lines 196-197: if, according to ANI, the two strains are the same species, why don’t you rename Asp10699 to Acaryochloris marina? And why do you refer to them as two species (e.g. in Figure legends)
Fig. 6. Specify again which color refers to which strain
Author Response
Thank you for spending your valuable time to peer review our paper. I believe our paper got improved based on the points suggested by you.
I respond to your points as follows. (our answers are in blue text.)
Α. sp. : It not acceptable to abbreviate the genus when it followed by sp. Please correct it as Acaryochloris sp. throught the manuscript, for clarity.
>We have revised as you mentioned.
lines 85-91: please explain the NBRC abbreviation; please give details on the isolation area and the nature of the environment it was isolated from (if details are available); please give the wavelength of FRL40SW
> We added the explanation of NBRC as follows.
Acaryochloris marina MBIC10699 was purchased from Biological Resource Center, NITE (NBRC; Japan) as Acaryochloris sp. NBRC102871.
This strain was isolated from the sea area of Palau, where it was in a symbiotic state with ascidians. This information will be published on the DDBJ database. FRL40SW is a normal white fluorescent lamp. we added a note in the context that it is a white fluorescent lamp.
lines 99-100: please specify which chl absorbs in which wavelength, for clarity
>Chl a was detected by 440 nm and Chl d was detected by 690 nm. To clarify the explanation, we revised as follows
HPLC analysis (Shimadzu LC20-AD) was performed according to Zapata’s method [11] and Chl a and d were detected by absorption at 440 nm and 690 nm, respectively.
line 168 and elsewhere: please make sure italics are used for taxa, chl, and genes
>It was a mistake that happened when copy-pasting the text. We have checked and corrected the mistakes throughout the text.
lines 196-197: if, according to ANI, the two strains are the same species, why don’t you rename Asp10699 to Acaryochloris marina? And why do you refer to them as two species (e.g. in Figure legends)
>As you indicated, we have renamed Acaryochloris sp. MBIC10699 to Acaryochloris marina MBIC10699. The description of two species was changed to two strains.
Fig. 6. Specify again which color refers to which strain
>We added the following explanation.
The plasmids shown in red are from A. marina MBIC10699, and the plasmids shown in blue are from A. marina MBIC11017.
Round 2
Reviewer 1 Report
The authors correctly clarified my comments and then implemented my suggestions in the new paper version. For this reason, I consider the paper accepted in the present form for the publication.